# Research on Temperature Dependence of Single-Event Burnout in Power MOSFETs

**DOI:** 10.3390/mi14051028

**Published:** 2023-05-11

**Authors:** Chen Wang, Yi Liu, Changqing Xu, Xinfang Liao, Dongdong Chen, Zhenyu Wu

**Affiliations:** 1Laboratory of Digital IC and Space Application, School of Microelectronics, Xidian University, Xi’an 710071, China; wdh8537@163.com (C.W.);; 2Guangzhou Institute of Technology, Xidian University, Guangzhou 510555, China

**Keywords:** power MOSFETs, temperature dependence, LET, single-event burnout (SEB)

## Abstract

Power MOSFETs are found to be very vulnerable to single-event burnout (SEB) in space irradiation environments, and the military components generally require that devices could operate reliably as the temperature varies from 218 K to 423 K (−55 °C to 150 °C); thus, the temperature dependence of single-event burnout (SEB) in power MOSFETs should be investigated. Our simulation results showed that the Si power MOSFETs are more tolerant to SEB at a higher temperature at the lower LET (10 MeV∙cm^2^/mg) due to the decrease of the impact ionization rate, which is in good agreement with the previous research. However, the state of the parasitic BJT plays a primary role in the SEB failure mechanism when the LET value is greater than 40 MeV∙cm^2^/mg, which exhibits a completely different temperature dependence from that of 10 MeV∙cm^2^/mg. Results indicate that with the temperature increasing, the lower difficulty to turn on the parasitic BJT and the increasing current gain all make it easier to build up the regenerative feedback process responsible for SEB failure. As a result, the SEB susceptibility of power MOSFETs increases as ambient temperature increases when the LET value is greater than 40 MeV∙cm^2^/mg.

## 1. Introduction

Power MOSFETs are widely used in space electronic systems, attributed to their strong driving ability, low power consumption, and high block voltages [1,2]. However, power MOSFETs are also found to be very vulnerable to single-event burnout (SEB) in a space irradiation environment, which is a catastrophic failure and can be triggered by the energetic particle penetrating the device in the off-state [3,4,5].

It has been extensively proven that the SEB failure of power MOSFETs is related to the establishment of the regenerative feedback process, where the two relevant mechanisms are the current amplification of the parasitic BJT and the carrier multiplication at the N-epi/N+sub junction [3,6]. That is, the turn-on and amplification of the parasitic BJT can be maintained with the hole current provided by the impact ionization process, meanwhile, the carrier impact ionization at the N-epi/N+sub junction can also be fed by the electron current coming from the parasitic BJT. This continuous reaction can ultimately lead to the catastrophic and permanent thermal failure of power MOSFETs. Nowadays, the Technology Computer-Aided Design (TCAD) simulator can help us know more details about the SEB failure in power MOSFETs and study other radiation-hardened techniques easily [7,8,9].

As we know, the ambient temperature varies dramatically in space flight missions; therefore, we have to consider the influence of the ambient temperature on the radiation tolerance of power MOSFETs. Some research has revealed that the SEB tolerance of power MOSFETs increases with the increase of ambient temperature, that is, the worst case for SEB occurs at a lower temperature [10,11,12,13]. This is due to that the impact ionization rate decreases with increases in temperature. However, on the other hand, the increase in temperature will increase the forward current of the emitter junction of parasitic BJT, which makes it easier to form a local hot spot and induce the SEB failure. Therefore, the two factors influenced by ambient temperature exhibit a completely opposite effect on the SEB performance of power MOSFETs. Thus, to get a better understanding of the temperature dependence of the SEB failure in power MOSFETs, we have to find out which factor plays a leading role in the different conditions.

In this paper, we find that for power MOSFETs, the SEB failure has an opposite temperature dependence at the lower and higher LET. At the lower LET (10 MeV·cm^2^/mg), the SEB susceptibility of power MOSFETs decreases with the increasing temperature since the decrease of the impact ionization rate plays a dominant role in this case, which agrees well with the previous research [10,11,12]. While, at the higher LET (100 MeV·cm^2^/mg), the condition is just the opposite. With the increase in temperature, the transient response of power MOSFETs to the high-energy particle strike is stronger. This is because the parasitic BJT starts to play a primary role in the regenerative feedback mechanism now. Due to the lower difficulty to turn on the parasitic BJT and the increasing current gain at a higher temperature, the worst case for SEB occurs at a higher temperature at the higher LET.

## 2. TCAD Simulation Model

The schematic cross-sectional view of the simulated power MOSFET device is shown in Figure 1. Table 1 gives the corresponding device structure parameters in detail. As shown in Figure 2, the breakdown voltage (BV) and threshold voltage (V_TH_) of the device are 424 V and 5.5 V, respectively, which are in good agreement with [7,14]. The single event burnout (SEB) threshold voltage of the device in [14] is 330 V (50% BV) under a LET value of 17 MeV∙cm^2^/mg, and the simulated device in this paper has a SEB threshold voltage of 307 V (72% BV) under the same LET value, exhibiting an increase of about 20% since the adoption of the SEB hardening techniques of the Pplus extension and the insertion of the buffer layer in our device structure [15,16,17].

The numerical simulations are performed using the Sentaurus TCAD simulator in this paper. Several physical models have been used to simulate the SEB failure process, including (1) the mobility degradation model, which contains ionized impurity scattering, carrier–carrier scattering, interface scattering, and high field velocity saturation. (2) recombination model, which includes Shockley Read Hall (SRH) recombination considering doping concentration and temperature dependence, Auger recombination regarding the high carrier densities caused by the heavy ion impact. (3) University of Bologna impact ionization model, which is developed to model the carrier multiplication at the N-epi/N+sub junction. (4) electro-thermal model, which considers the lattice temperature variation induced by the large current density after a heavy ion’s strike by adding the thermodynamic model to simulate the lattice self-heating effect. In this paper, we define the substrate contact as an ideal heat sink, and its temperature is fixed at the ambient temperature.

Moreover, the heavy ion model is activated to simulate the transient response of the power MOSFET to the ion. The carrier generation rate induced by the heavy ion is computed by:(1)$$G\left(l,\omega ,t\right)=G_{LET}\left(l\right)R\left(\omega ,l\right)T\left(t\right)$$
where *l* is the length of the particle strike track, *ω* is the radius defined as the perpendicular distance to the track and t is the transient time. In addition, *R*(*ω*,*l*) and *T*(*t*) are Gaussian functions describing the spatial and temporal variations of the carrier generation rate with a characteristic radius of *w*_0_ and a characteristic time of *S_hi_*, respectively. The LET is assumed to be a constant value when the energetic ion passes through the device. In this work, the particle strike track is perpendicular to the surface of the device, and the track length is set to 80 μm, which is able to penetrate the entire device. Table 2 shows the default heavy ion parameters used for our simulations.

## 3. Results and Discussion

### 3.1. Single Event Burnout Sensitive Region

Previous investigations have shown that the neck region of power MOSFETs is the sensitive area to SEB [8,12,18,19]. This is because the neck region, which is far from P-body, is more resistive when the lateral hole current flows to the source electrode, developing a larger ohmic voltage drop across the emitter-base junction, making the parasitic BJT easier to be forward biased. Thus, the corresponding neck region from *x*_0_ = 31 μm to *x*_0_ = 50 μm of the simulated power MOSFET is chosen to obtain the most sensitive ion strike position in our simulations. Figure 3 exhibits the relationship between the ion’s incident position *x*_0_ and SEB threshold voltage (*V_th,SEB_*) at LET = 75 MeV∙cm^2^/mg, where *V_th,SEB_* is defined as the minimum drain voltage required to trigger SEB. It can be concluded that *x*_0_ = 40 μm is the most sensitive incident position with the lowest *V_th,SEB_* of 164 V. Therefore, *x*_0_ = 40 μm is selected as the default incident position in our following simulations so as to obtain the worst case for SEB. In addition, it should also be pointed out that we regard the SEB failure criterion as the maximum lattice temperature of the device after the ion strike reaches the melting point of silicon (1688 K) [9,20].

### 3.2. The Temperature Dependence of SEB at Different LETs

Figure 4 shows *V_th, SEB_* as a function of ambient temperature (Ta) at different LETs. Our selected ambient temperature varies from 218 K to 423 K (−55 °C to 150 °C), which corresponds to the operating temperature range of military components. It is obvious that the SEB response exhibits different temperature dependencies at different LET values. At the lower LET value of 10 MeV∙cm^2^/mg, *V_th,SEB_* increases slightly with the increase of ambient temperature. On the contrary, with the LET value greater than 40 MeV∙cm^2^/mg, *V_th,SEB_* decreases as temperature increases. To get a better understanding of the discrepancy of the temperature dependence of SEB at different LETs, the following discussions are mainly focused on the two conditions: condition A is at the lower LET (10 MeV∙cm^2^/mg) and Vds = 300 V; condition B is at the higher LET (100 MeV∙cm^2^/mg) and Vds = 100 V.

#### 3.2.1. The Temperature Dependence of SEB at the Lower LET

In order to analyze the influence of ambient temperature on SEB sensitivity at the low LET, Figure 5 demonstrates the impact ionization rate distributions at Ta = 250 K and Ta = 423 K under condition A (LET = 10 MeV∙cm^2^/mg, Vds = 300 V). From Figure 5, we can find that the peak impact ionization rate is located at the P-body/N-epi junction after the ion strike 100 ps, while it shifts to the N-epi/N-buffer interface after the ion strike 10 ns. This is due to the Kirk effect caused by the injection of electrons from the emitter to the collector and the resulting less positive total charge density in the collector region [18,21]. At Ta = 250 K, the peak impact ionization rates are 9.71 × 10^25^/cm^3^·s and 6.08 × 10^25^/cm^3^·s after the ion strike 100 ps and 10 ns, respectively. However, when Ta = 423 K, the peak impact ionization rates are 6.68 × 10^25^/cm^3^·s and 4.06 × 10^25^/cm^3^·s, respectively, which decreases about 40% with the increase of Ta from 250 K to 423 K. This is because the increase of temperature will aggravate the vibration of the lattice, which will lower the mean free path of the carriers and decrease the impact ionization rate in the device. Therefore, the transient current induced by the carrier multiplication process will be smaller at a higher ambient temperature. Consequently, the maximum lattice temperature after the ion strike at Ta = 423 K (826 K) decreases compared with the case at Ta = 250 K (928 K) as shown in Figure 6. Our simulations show that the SEB tolerance of power MOSFETs has a positive correlation with the ambient temperature at the lower LET, which is in good agreement with the literature report [10,11,12].

#### 3.2.2. The Temperature Dependence of SEB at the Higher LET

Then, the condition at the higher LET (condition B) is discussed. Figure 7 shows the lattice temperature and drain current variations of the device versus time at different Ta under condition B (LET = 100 MeV∙cm^2^/mg and Vds = 100 V). It can be seen that the ambient temperature has an important influence on the SEB performance of power MOSFETs. With Ta increasing, the maximum lattice temperature after the ion strike varies from 604 K (Ta = 250 K) to 1010 K (Ta = 350 K). These values do not exceed the melting point of silicon and the lattice temperature can eventually recover to the ambient temperature, indicating no burnout. However, at Ta = 398 K, the maximum lattice temperature rises rapidly and can even exceed 1688 K, leading to catastrophic failure, and the drain current can maintain a high level. Once the SEB failure criterion is triggered, the simulation is terminated, and thus the simulation stops at about 100 ns at Ta = 398 K. From the transient temperature response, it can be observed that the device is more sensitive to SEB at a higher temperature under condition B, completely different from which we observed under condition A.

To find out the reason why the temperature dependence of SEB differs at the different LETs, the electric field distributions at the lower LET and higher LET are comparatively studied in Figure 8. As we can see, because of the Kirk effect after the ion strike, the maximum electric field will move from the P-body/N-epi junction to the N-epi/N+buffer interface (N-buffer/N+sub interface) as time increases. However, the fewer generated charges at the lower LET cannot maintain the peak electric field at the N-epi/N-buffer interface and the maximum electric field returns to the P-body/N-epi junction after the ion strike 5 μs. In contrast, at the higher LET, it will be easier to maintain the peak electric field at the N-buffer/N+sub interface due to the more deposited charges in the epitaxy layer [16]. Therefore, at the higher LET, whether the SEB will occur or not mainly depends on the state of the parasitic BJT, that is to say, whether the electron current provided by the amplification of parasitic BJT can sustain the impact ionization process to build up the regenerative feedback. As a result, the influence of temperature on parasitic BJT plays a dominant role in SEB failure under condition B.

Figure 9 shows the electron current density distributions near the parasitic BJT at (a) Ta = 250 K and (b) Ta = 423 K under condition B. It can be seen that the electron current density near the collector region increases greatly as time increases. This is because the heavy ion-induced hole current moves laterally through the P-body to be collected by the source electrode, resulting in a forward voltage drop across the emitter junction to turn on the parasitic BJT and amplify the current. Compared with the case at Ta = 250 K, the electron current density at Ta = 423 K is larger, especially after 150 ps. We assume that the parasitic BJT is turned on when the ratio of the source electron current to source total current reaches 0.97 in our simulations, while it indicates that the emitter junction is turned on and the electron current has become the dominating part of the source total current. For Ta = 423 K, it only takes 99.2 ps to turn on the parasitic BJT while it takes 225.8 ps for Ta = 250 K. Thus, it can be concluded that the parasitic BJT at Ta = 423 K is turned on at about 100 ps earlier than that at Ta = 250 K. This is because the ambient temperature has a significant influence on the built-in voltage of the emitter-base junction and the base resistance (R_B_). With the increasing temperature, the intrinsic carrier concentration will increase exponentially, reducing the built-in potential of the emitter-base junction, meanwhile, R_B_ increases caused by the decreased mobility, both making the parasitic BJT easier and earlier to be turned on [22,23].

Besides, we know that the lifetime of the carriers will increase with rising temperature, and it can be modeled by a power law [24,25]:(2)$$\tau \left(T\right)=\tau_{0}{\left(\frac{T}{300\;K}\right)}^{\alpha }$$
where *τ*_0_ is the carrier lifetime when the lattice temperature is 300 K, *T* is the lattice temperature and *α* is 2.1 [26]. Figure 10 shows a comparison of the minority carrier electron lifetime distribution in the base region of the BJT at Ta = 250 K and Ta = 423 K, from which we can observe the longer minority carrier electron lifetime at Ta = 423 K and it is consistent with the relationship between carrier lifetime and temperature as shown in (2). Thus, with the higher ambient temperature, more electrons injected from the emitter can be collected by the reverse-biased collector junction, resulting in a larger current gain.

The current gain β of the parasitic BJT in the power MOSFET cannot be obtained directly because of the unmeasurable base current I_B_. So, we calculate the common-base current gain α of the parasitic BJT firstly by extracting the ratio of the drain current (collector current) to the source current (emitter current) of the power MOSFET. Then we get the common-emitter current gain β of the BJT through the relationship between α and β as shown in (3). Table 3 gives the calculation results of the current gain of the parasitic BJT under condition B. As we can see, in the whole temperature range, the current gain varies from 19.44 to 123.24, increasing by a factor of five to six. The current gain of the BJT increases strongly with the increasing temperature, which is responsible for the large drain current and higher temperature rise at higher ambient temperature in Figure 7.
(3)$$\beta =\frac{\alpha }{1-\alpha }$$

In conclusion, with the LET value increasing to 40 MeV∙cm^2^/mg, the temperature dependence of SEB in power MOSFET is completely different from that of 10 MeV∙cm^2^/mg. Due to the more deposited charges in the epitaxy region, it will be easier to maintain the peak electric field at the N-epi/N-buffer junction (N-buffer/N+sub interface), therefore, whether the parasitic BJT can provide continuous and enough electron current for the carrier multiplication process becomes extremely important. Our research shows that with the ambient temperature increasing, the difficulty to turn on the parasitic BJT decreases due to the lower built-in potential and the increasing base resistance, meanwhile, the current gain increases due to the increasing minority electron lifetime in the base region [27,28,29,30]. As a result, it is easier to form a local hot spot and induce the SEB failure at a higher temperature, thus power MOSFET is more sensitive to SEB at a higher temperature when the LET value is greater than 40 MeV∙cm^2^/mg.

## 4. Conclusions

This paper presents the simulation-based comparison of the temperature dependence of single-event burnout (SEB) for power MOSFETs at different LETs. Our simulations show that at the lower LET (10 MeV∙cm^2^/mg), with the ambient temperature (ranging from −55 °C to 150 °C) increasing, the peak impact ionization rate decreases, resulting in stronger robustness to SEB at a higher temperature. However, the temperature dependence of SEB exhibits different responses when the LET value increases to 40 MeV∙cm^2^/mg. This is because the more deposited charges in the collector region make it easier to maintain the carrier multiplication thus the state of the parasitic BJT starts to play a dominant role in the regenerative feedback mechanism. When the LET is 100 MeV∙cm^2^/mg, the time to turn on the parasitic BJT reduces from 225.8 ps to 99.2 ps, and the current gain increases from 19.44 to 123.24 due to the increase of the minority lifetime in the base region with the temperature increasing from 250 K to 423 K, which will be easier to form a local hot spot and induce the SEB failure at a higher temperature. Consequently, the power MOSFET exhibits stronger SEB robustness at a lower temperature at the higher LET.

## Figures and Tables

**Figure 1 micromachines-14-01028-f001:**
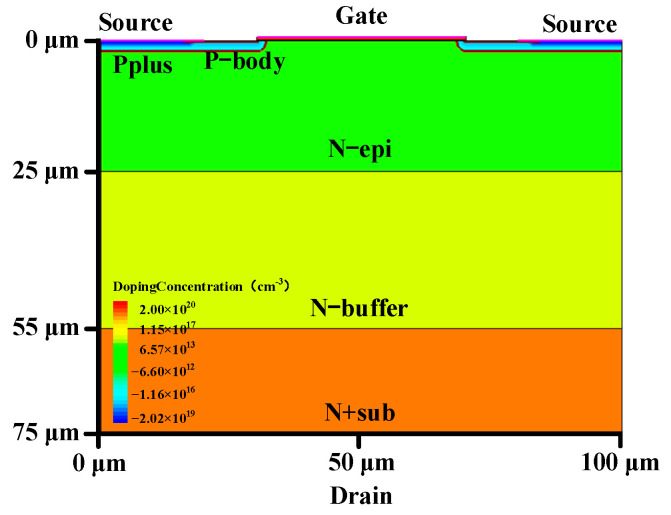
Schematic cross-sectional view of the simulated power MOSFET.

**Figure 2 micromachines-14-01028-f002:**
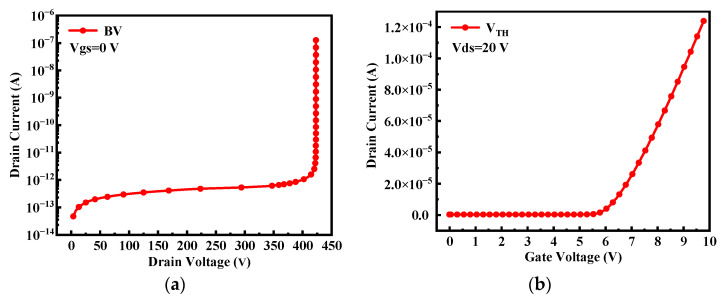
Static characteristics of the power MOSFET device (**a**) breakdown characteristic and (**b**) transfer characteristic.

**Figure 3 micromachines-14-01028-f003:**
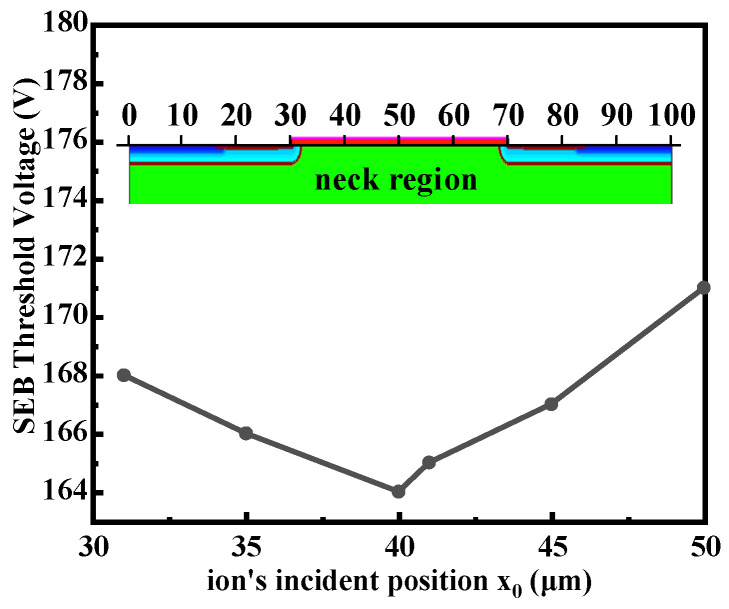
SEB threshold voltage as a function of the ion strike position (LET = 75 MeV∙cm^2^/mg).

**Figure 4 micromachines-14-01028-f004:**
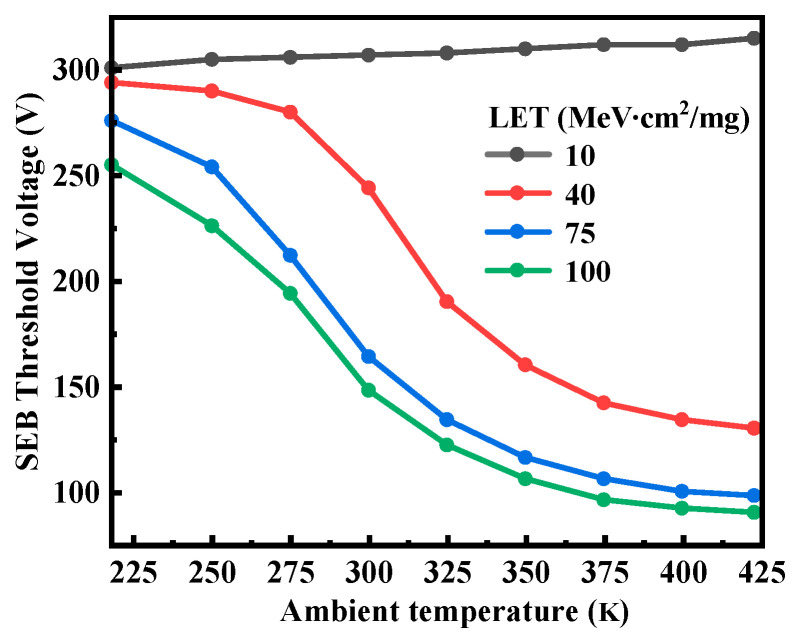
Relationship between *V_th, SEB,_* and ambient temperature as a function of LET.

**Figure 5 micromachines-14-01028-f005:**
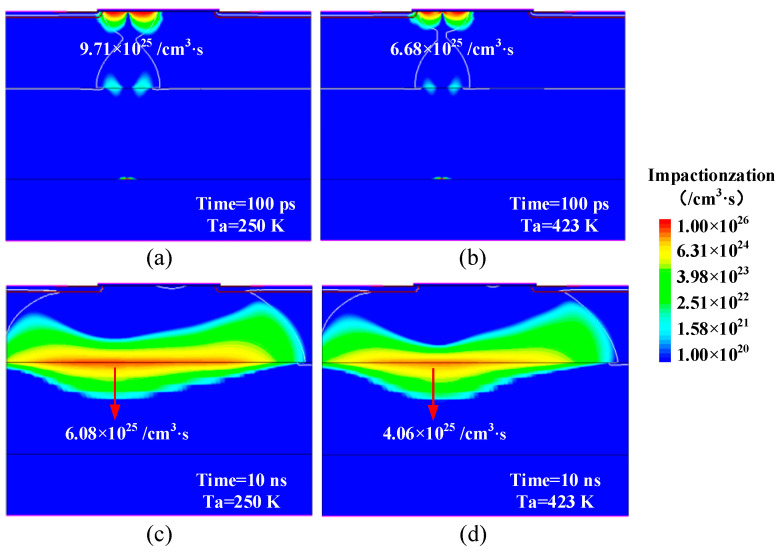
The impact generation rate distributions of the device at (**a**) Ta = 250 K, (**b**) Ta = 423 K, (**c**) Ta = 250 K, (**d**) Ta = 423 K after the ion strike under condition A (LET = 10 MeV∙cm^2^/mg, Vds = 300 V).

**Figure 6 micromachines-14-01028-f006:**
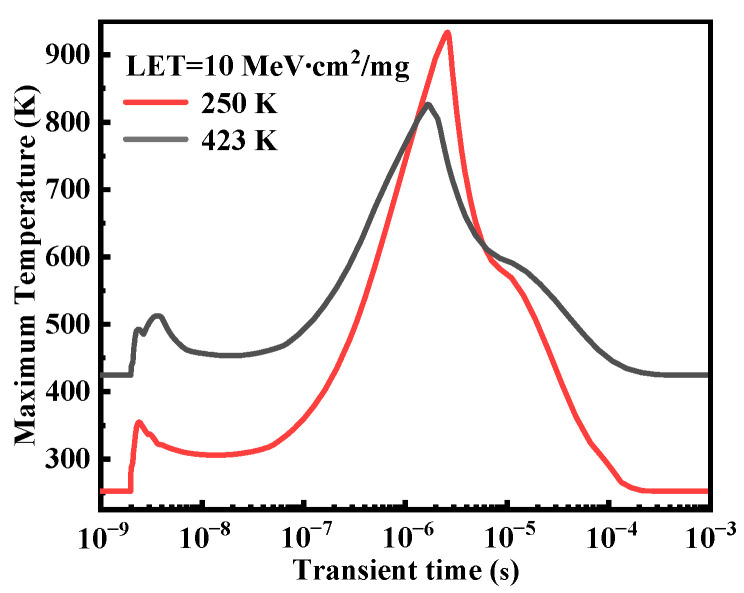
The temperature response after the ion strike under condition A (LET = 10 MeV∙cm^2^/mg and Vds = 300 V).

**Figure 7 micromachines-14-01028-f007:**
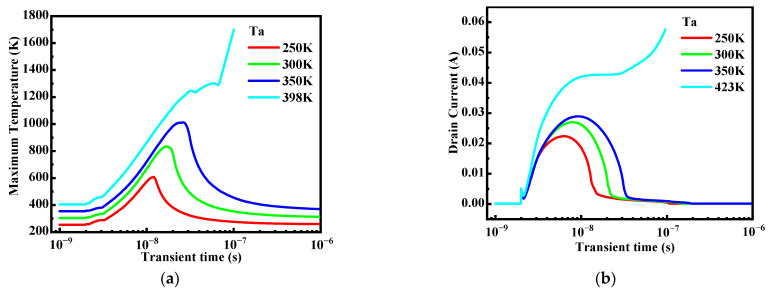
The SEB response of the power MOSFET at different Ta under condition B (LET = 100 MeV∙cm^2^/mg and Vds = 100 V) (**a**) temperature responses and (**b**) drain current responses.

**Figure 8 micromachines-14-01028-f008:**
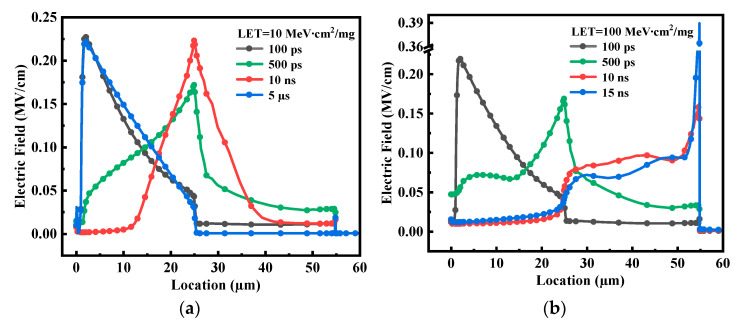
The electric field distributions of power MOSFET after an ion’s strike at (**a**) LET = 10 MeV∙cm^2^/mg and (**b**) LET = 100 MeV∙cm^2^/mg (Ta = 250 K, Vds = 300 V).

**Figure 9 micromachines-14-01028-f009:**
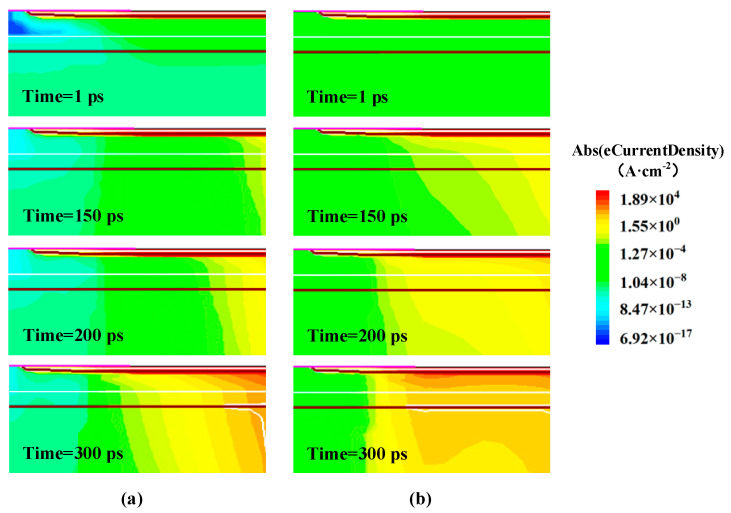
The electron current density distributions near the parasitic BJT after the ion strike at (**a**) Ta = 250 K and (**b**) Ta = 423 K under condition B.

**Figure 10 micromachines-14-01028-f010:**
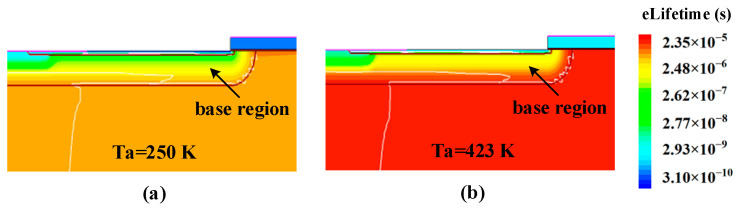
The minority carrier electron lifetime distribution in the base region under condition B after the ion strike 500 ps at (**a**) Ta = 250 K and (**b**) Ta = 423 K.

**Table 1 micromachines-14-01028-t001:** Device structural parameters for simulations.

Device Parameters	Values
Gate Oxide Thickness (nm)	100
N+ Source Doping (cm^−3^)	2 × 10^20^
Pplus Doping (cm^−3^)	2 × 10^19^
P-Body Doping (cm^−3^)	2 × 10^17^
N-Drift Thickness (µm)	25
N-Drift Doping (cm^−3^)	5 × 10^14^
N Buffer Thickness (µm)	30
N Buffer Doping (cm^−3^)	5 × 10^15^

**Table 2 micromachines-14-01028-t002:** Parameters used for the heavy ion simulations.

Track Length *l*_0_	Track Radius *w*_0_	Characteristic Time of Gaussian Function *S_hi_*	Peak Charge Generation Time *t*_0_	Ion Incident Position *x*_0_
80 μm	0.02 μm	4 ps	2 ns	40 μm

**Table 3 micromachines-14-01028-t003:** The current gain of the parasitic BJT of power MOSFET as a function of Ta under condition B after the ion strike 500 ps.

Ta	β_(BJT)_
250 K	19.44
300 K	24.95
350 K	40.49
400 K	82.27
423 K	123.24

## Data Availability

Not applicable.

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
