# Peer review of "Research on Temperature Dependence of Single-Event Burnout in Power MOSFETs"

_micromachines, 2023, doi:10.3390/mi14051028_

Round 1

Reviewer 1 Report

C. Wang et al. present the simulation-based comparison about the temperature dependence of single-event burnout for power MOSFETs at different LETs. Th manuscript is informative and well organized. Th reviewer's comments are below:

1) It seems that the authors have used the Sentaurus TCAD simulator. They should give some references linked to the study and investigation of single-event burnout in power MOSFETs via the Sentaurus TCAD simulator.

2) A validation of the simulator against some experimental data will be of great benefit and will confirm the recorded results.

3) In Fig. 7, I would enquire about the continuity of blue curve (Ta 398K).

Good luck 

Reviewer 2 Report

I recommend the paper for publication after some changes. I believe that the conclusions of this paper are somewhat already known by other researchers working in this field. The work presented here may add to this common knowledge; however, I do not believe that this is new. If the authors had supported their findings with experiments, it would have been a significantly stronger paper.

Please proofread the paper, and especially the abstract section. Here is an example:

“with the temperature varies from” => “as the temperature varies from”

Figure 1: The neck region of the silicon power device is very large, and so is the cell size. Is this a realistic cell structure?

Why does the SEB threshold reach its lowest value at 40 micrometers? There is nothing fundamentally different at 40 and 60, if the device symmetry is taken into account. Please explain why the middle of the neck region is not the most sensitive location.

Reviewer 3 Report

The authors through TCAD simulations investgate the temperature dependence of the SEB faliure in power MOSFETs due to energetic particles penetrating the device. They conclude that at low values of the linear charge deposition the probability of SEB event decreases with temperature increasing, whereas at high values of the deposited charge the behaviour with temperature changes in the opposite way. Although the first case is confirmed by experimental data, the stronger SEB robustness at low temperatures and at the high ion LET it seems that is not supported by experimental data. Although the SEB temperature dependence issue is interesting for the estimation of the reliability of these devices in applications, the authors should clarify the reasons why they find this result interesting which seems to be limited to simulations. In my opinion, the manuscript can be published after  a minor revision

Abstract

Power MOSFETs are found to be very vulnerable to single-event burnout (SEB) in space irradiation environment

Power MOSFETs are subject to SEB failures due also to the atmospheric neutrons. Hence, the reliability of Power MOSFETs used in automotive applications must take into account the SEB failures due to atmospheric neutrons

It has been extensively proven that the SEB failure of power MOSFETs is related to the establishment of the regenerative feedback process, where the two relevant mechanisms are the current amplification of the parasitic BJT and the carrier multiplication at the N-epi/N+sub junction

This mechanism is valid for silicon power MOSFETs. For SiC MOSFETs this mechanism to explain SEB faliure is questionable (see for example Lichtenwalner et al. “ Reliability of SiC Power Devices against Cosmic Ray Neutron Single-Event Burnout”,  Materials Science Forum doi:10.4028/www.scientific.net/MSF.924.559)

The authors should stress that manuscript is focused on silicon devices.

we have to find out which factor plays a leading role at the different conditions.

It is not clear for me what the different conditions refer to.

In this paper, we find that for power MOSFETs, the SEB failure has an opposite temperature dependence at the lower and higher LET

The authors do not mention the type of interaction that takes place in the device (alpha particles, protons, ions) and refer only to the LET of the ionizing particle. This quantity is expressed in MeV cm^2/g unit. You consider throughout the manuscript the linear charge deposition (LCD) measured in (pC/mm) unit which is different from LET. You have to clarify the relationship between these quantities.

Line 95: . And the LET

Lines 117-119:  we regard the SEB failure criterion as the maximum lattice temperature of the device after the ion strike reaches the melting point of silicon (1688 K) [19,20]

 In [19] the silicon melting point is 1700 K. In [20] just the critical current is considered.

Lines 141 and 142: Figure. 5

Why the dot?

Line 150: about 30% with the increase of Ta

about 40%

Lines 183-185: The maximum electric field at the N-buffer/N+sub interface can reach 0.39 MV/cm, which allows the establishment of carrier multiplication process.

This value of the electric field is over that of the breakdown field in silicon. The carrier multiplication process occurs even at lower values of the electric field.

Lines 199-200 Figure 9: Compared with the case at Ta=250 K, the electron current density at Ta=423 K is larger, especially after 100 ps.

The colors do not allow to distinguish the difference in intensity of the current density between Figure (a) and (b). At the lower temperature and after 100 ps is observed that the region with high current density is wider than that at high temperature. 

Round 2

Reviewer 2 Report

I still do not believe the neck region width is reasonable. Ref [1] does not show a device with a neck region of 40 micrometers as the authors say. Also Ref [1] is from ~40 years ago. This is a huge device that is not practical. Some of the conclusions would be affected by the device design. That being said, I still believe that the paper might be of interest to some researchers.  
